# Preschoolers’ Perceptual Analogical Reasoning and Map Reading: A Preliminary Study on the Mediating Effect of Spatial Language

**DOI:** 10.3390/children10040630

**Published:** 2023-03-28

**Authors:** Marco Giancola, Maria Chiara Pino, Valentina Riccio, Laura Piccardi, Simonetta D’Amico

**Affiliations:** 1Department of Biotechnological and Applied Clinical Sciences, University of L’Aquila, 67100 L’Aquila, Italy; 2Department of Psychology, Sapienza University of Rome, 00185 Rome, Italy; 3San Raffaele Cassino Hospital, 03043 Cassino, Italy

**Keywords:** perceptual analogical reasoning, map reading, navigation, spatial language, children, topographical disorientation, cognitive rehabilitation, mediation

## Abstract

Reading and interpreting a map represents an essential part of daily life, enabling appropriate orientation and navigation through space. Based on the idea that perceptual analogical reasoning is critical in aligning the spatial structure of the map with the spatial structure of the space and given the critical role of language, especially spatial language, in encoding and establishing spatial relations among elements in the environment, the present study investigated the joint contribution of perceptual analogical reasoning and spatial language in map reading. The study was conducted with 56 typically developing 4- to 6-year-old children, and the results indicated that perceptual abstract reasoning affected map reading through the mediating effect of spatial language. These findings yielded theoretical and practical implications regarding the role of perceptual abstract reasoning and spatial language in shaping map-reading abilities in the early stages of life, highlighting that domain-specific language competencies are necessary to improve the encoding of spatial relations, to establish object correspondences, and to ensure successful navigation. Limitations and future research directions were discussed.

## 1. Introduction

Human beings have been defined as a “symbolic species” due to their ability to process, learn, and use a rich array of symbol systems, such as language, pictures, and signs [1]. For instance, topographic maps represent an essential symbol-based system of reality, characterized by data points that are useful for representing, communicating, and guiding navigation through the surrounding environment [2]. All animals, from ants to primates, have developed different mechanisms to navigate the environment, but only humans can support their navigation by reading topographic maps to reach a given location [2]. Maps can be described as a complex communication navigational system as well as a precious tool to graphically represent spatial information. Humans’ ability to read a map (map reading) is widely defined as a basic aspect of map use, which entails extracting information about the surrounding environment, finding locations, and recognizing symbols and their meanings [3,4]. According to Siegel and White’s model [5], map-like representation is the last stage of environmental knowledge acquisition because it prescinds from the egocentric perspective and asks the individual to take an allocentric perspective. In this vein, a long research tradition suggests that map reading emerges relatively late during childhood. Three-year-old children were found to be able to use a map to locate an object in a room using visual matching, motor priming [6], and local cues to the target location, for instance, when the figures depicted in the map were visually similar to their referents [7]. By four years of age, children use their map-reading competencies to navigate small-scale environments (i.e., a single room) by detecting geometric correspondence between three-dimensional layouts and two-dimensional maps and using such correspondences to guide navigation [2]. Furthermore, 4- and 5-year-olds can reconstruct an array of objects in a room from memory after memorizing the arrangement depicted on simple maps [8]. Nevertheless, other research has stressed that map reading seems to emerge before three years of age. Children between 24 and 30 months were found to use a photograph as a cue to the location of a hidden object in a picture-based retrieval game and successfully navigate small-scale environments using a map, e.g., [2,9]. However, Winkler-Rhoades and colleagues [10] found that 2.5-year-old children spontaneously used geometric information from 2D maps to locate objects in a 3D surface layout, even though the maps differed from the layouts in terms of size, mobility, orientation, dimensionality, and perspective.

Overall, interpreting a map requires the conceptualization of space in terms of distance between objects, distribution and exact location of objects, and the involvement of high-order cognitive abilities (e.g., mental rotation, planning, and reasoning), which allow for thinking about a spatial scenario [11] and ensure successful navigation in terms of wayfinding, building recognition, spatial reorientation, and moving throughout a new town [12,13].

Classical research suggests that using spatial competencies, such as map reading, is a more basic ability than analogical reasoning [14,15]. The latter represents a core element of human cognition, which involves recognizing and comparing different objects, events, or situations, as well as determining what they have in common [16]. This type of reasoning represents a critical factor of children’s higher-order cognitive system, enabling them to make inferences about novel phenomena, transfer learning across contexts, and extract relevant information from everyday experiences through relational similarities [17]. Differently from the Piagetian perspective, stressing that children are unable to reason by analogies prior to achieve the formal operation stage (13–14 years of age), the relational primacy hypothesis [18,19,20] posits that analogical reasoning is basically available since the early stages of life and grows with age due to the accretion of knowledge. For instance, Goswami et al. [21] found that children as young as 4 years were able to solve perceptual analogical tasks in which two physical causal relations were manipulated to change an object. Interestingly, past research has suggested that children’s perceptual analogical reasoning is critical in understanding and using graphical spatial representations because it involves finding similarities or correspondences between two elements and transferring the information to an analogous scenario [22]. In this vein, Yuan et al. [23] argued that perceptual analogical reasoning is centrally involved in connecting a map with its referent space. The authors found that preschoolers used their analogical reasoning skills to align the relative spatial structure of a map with the relative spatial structure of the space [23]. Although these findings call for a role of young children’s analogical reasoning in map reading, this topic remains relatively underexplored. This is probably due to a long tradition suggesting that children under 11 or 12 years show fragile analogical reasoning depending on the ability to ignore irrelevant perceptual distractors, limited knowledge about the world, and the immaturity of neural networks [24,25].

Interestingly, analogical reasoning seems to be associated not only with map reading but also with language abilities. In particular, even though some studies have found that language and analogical reasoning have a reciprocal association [26], others have stressed that analogical reasoning could trigger different aspects of language, including the ability to learn syntax, new grammatical constructs, the meaning of adjectives and verbs, and the ability to learn categories about objects. This evidence has been corroborated mainly by research on children with developmental language disorders, in which language delay has been viewed as the result of problems with nonverbal reasoning, e.g., [27]. For instance, Nippold and colleagues [28] found that children with language disorders had poorer analogical performance than their age-matched peers in three analogy tasks. However, these differences disappeared when analogical performances were controlled between groups. Similarly, Leroy and colleagues [29,30] argued that deficits in linguistic and nonlinguistic (perceptual) analogical reasoning represent one of the leading causes of language disorders in children with specific language impairment. Overall, these findings offer theoretical grounds to reasonably assume that analogical skills play an essential role in language capabilities.

Furthermore, a groundswell of academic research suggested that language capabilities trigger spatial skills and that this effect is mainly domain-specific instead of domain-general [31,32]. Indeed, spatial language, which is a specific kind of language consisting of words and phrases typically used to describe the spatial relationships between elements [33], positively affects the development of spatial skills, e.g., [34,35,36,37,38,39,40]. Using spatial language can affect how people represent and reason about space, e.g., [38,41]. For instance, Loewentein and Gentner [38] showed that hearing words that name spatial relations facilitated children’s encoding and mapping of spatial relations. Moreover, the literature suggests that human representations of space and its development are affected by how space is codified in the language people learn [32,38]. Further evidence of the relationship between spatial language and spatial ability, such as navigational ability, is that both skills allow individuals to mentally and physically organize objects in their world [42,43]. Piccardi and colleagues’ research [42] highlighted that children’s verbal comprehension of spatial locatives (e.g., in front of/behind, from/to, below/above, up/down) predicted the ability to learn, retrieve, and graphically represent pathways on a map of the environment. Furthermore, Gentner et al. [32] argued that spatial language facilitates spatial reasoning, suggesting that this specific kind of language contributes to forming conceptual representations of space. Overall, these findings confirm that spatial language is necessary to develop navigational skills [42].

### The Present Study

The current research sought to deepen the association between perceptual analogical reasoning and map reading, also addressing the involvement of spatial language in a sample of typically developing 4- to 6-year-old children. Based on evidence that map reading depends on the acquisition of specific aspects of language [41], which, in turn, are triggered by perceptual analogical reasoning [14,15,22], we have advanced the existence of a relationship between analogical reasoning and map reading via language abilities. In this vein, we hypothesized a mediation model in which perceptual analogical reasoning is the focal predictor, map reading is the outcome, and language is the mediator. Since spatial language and map reading belong to the same domain (i.e., spatial ability) and given that the effect of language on spatial cognition is mainly domain-specific [31,32], we hypothesized that the perceptual analogical reasoning–map reading link was specifically fabled by spatial language. Specifically, spatial language should help children better encode spatial relations and establish object correspondences, thus ensuring successful navigation through the environment [32,38]. Therefore, the main hypothesis of the current research was formulated as follows: spatial language competencies mediate the association between perceptual analogical reasoning and map reading.

## 2. Materials and Methods

### 2.1. Participants

A sample of 56 preschoolers took part in this study (mean_age_ = 5.08 years; SD_age_ = 0.81 years; range_age_ = 4–6 years), of whom 38 (67.9%) were boys and the remaining 18 (32.1%) were girls. Of the 56 preschoolers, 16 were four years old (28.6%), 19 were five years old (33.9%), and 21 (37.5%) were six years old. All children were recruited from local kindergartens in L’Aquila, Abruzzo region (Italy), and were native Italian speakers. The study did not include foreign children and those with learning difficulties and other neurodevelopmental diseases (as reported by their teachers or families). None of the children had primary visual or hearing impairments, a diagnosis of neurological disorders, or any emotional or behavioural problems. Furthermore, none of the participants reported general language comprehension deficits, as captured by the Test for Reception of Grammar—version 2 [TROG-2]; [44], a standardized measure evaluating lexical, grammatical, and syntactic constructions of language.

### 2.2. Measures

#### 2.2.1. Perceptual Analogical Reasoning

The Coloured Progressive Matrices [CPM] [45] belong to a collection of standardized measures (Ravene’s progressive matrices—RPM) which was designed for individuals in the developmental stage. It comprises 36 items divided into three different series (12 items for each series), which depict graphic elements that change from left to right and from top to bottom and have a specific relation system. The graphic element is missing in the lower right part of the items (see Figure 1). Participants are asked to choose the element, one of six alternative elements presented below the main matrix, that best completes the matrix. The scores of the test ranged from 0 to 36, and the number of correct responses was collected. Notably, even though the CPM has been widely used to assess fluid intelligence [46], this kind of measure can be conceptualized as geometric analogy problems, in which the participant is requested to find a specific visual relation system embedded in a graphic matrix by applying analogical reasoning [22,47]. Therefore, in this research, we considered CPM to be a reliable instrument of perceptual analogical reasoning. This aligns with previous research suggesting that analogical reasoning lies at the heart of visual problem solving and intelligence more broadly [48].

#### 2.2.2. Spatial Language

The Test of Grammatical Comprehension for Children—2 [Test di Comprensione Grammaticale per Bambini, TCGB-2] [49] is a language comprehension test in which children had to choose pictures corresponding to target sentences uttered by the examiner, discriminating them among morphological-morpho-syntactical distractors. The TCGB is a picture multiple-choice language test composed of 76 sentences addressing different grammatical structures, such as locative (e.g., “The dog is above the chair”), active negative (e.g., “The girl doesn’t run”), passive negative (e.g., “The piano is not played”), passive affirmative (e.g., “The car is washed by the child”), and relative (e.g., “The vase that the child is painting is on the chair”). Given the goal of the current study, only the first 14 items on spatial locatives were considered to assess children’s spatial language. Notably, although the TCGB-2 score is usually provided by the number of wrong answers provided by the child, in this study, the number of correct responses were collected, with high scores reflecting high spatial language competencies.

#### 2.2.3. Map Reading

The Semmes Test [50] is a classic measure widely used to assess map reading in both children and adults and requires translating a schematic representation into locomotion. This test consists of five visual maps of five paths of increasing length and complexity. Each map was drawn on a 32 cm square card (see Figure 2), which schematically contains a drawing of a 3 × 3-point grid depicting 9 red dots (each 12 mm in diameter) laid out on the floor (3 × 3 m). The map contained a black-coloured pathway (4.5 mm in width), which participants had to follow without rotating the square card. A black circle around one of the dots indicates the starting point, and the arrowhead marks the end of the path. The nine red circles located on the floor were 15.5 cm in diameter and placed in a large room. The score of this test consists of the total sum of the correct paths performed by children.

### 2.3. Procedure

First, an interview was requested for school principals to provide a detailed explanation of the nature and the main procedures of the research. Second, parents or children’s legal tutors were sent a note detailing the main topic of the research, procedures, and assessment. Furthermore, the note clearly explained that children’s participation was voluntary and that anonymity was guaranteed. The research began after all parents or legal tutors provided their written consent. Children were tested individually in a quiet room of the kindergarten. This study relied on an extensive research project addressing cognitive functioning in children and was conducted in accordance with the Declaration of Helsinki. The research was ethically endorsed by the Research Ethics Committee of the University of L’Aquila (prot. n. 11/2020) on 21 April 2020.

### 2.4. Statistical Analysis

All analyses were performed by SPSS Statistics version 24 for Windows (IBM Corporation, Armonk, NY, USA). Descriptive statistics were used to analyse the demographic features of the sample, whereas correlational analysis was computed to preliminarily check the associations among study variables. Finally, the mediating effect of spatial language was evaluated by the PROCESS macro for SPSS (version 3.5) [51]. The significance of the mediating effects was analysed using 5000 resamples of bootstrapped estimates with 95% bias-corrected confidence intervals (CIs) [46]. Bootstrapping is a nonparametric approach that bypasses the problem of nonnormality and enables an accurate test of mediating [52] as well as moderating effects [53], mainly in small- to medium-sized samples [53,54,55,56,57]. The criteria for mediation indicate that the 95% CIs must not include zero [58]. Statistical significance was set to *p* < 0.05.

## 3. Results

The test for normality revealed that data were not normally distributed for spatial language (Kolmogorov–Smirnov Test: Z_TCGB_ = 0.001, *sig)* and map reading (Kolmogorov–Smirnov Test: Z_Semmes_ = 0.032, *sig*) and were normally distributed for perceptual analogical reasoning (Kolmogorov–Smirnov Test: Z_Raven_ = 0.589, *ns).* The z test revealed no outliers in the dataset, considering the range between −3.0 and +3.0 *z* scores as reference values. Furthermore, the preliminary Spearman correlation analyses showed that perceptual analogical reasoning was positively associated with spatial language (*r* = 0.37, *p* < 0.01) and map reading (*r* = 0.52, *p* < 0.01), and the latter was also positively associated with spatial language (*r* = 0.56, *p* < 0.01). Moreover, among demographics, only age was positively associated with perceptual analogical reasoning (*r* = 0.47, *p* < 0.01) and map reading (*r* = 0.49, *p* < 0.01). Table 1 reports descriptive statistics and correlations among all study variables.

Based on correlations, mediation analysis was performed by entering perceptual analogical reasoning as the independent variable and map reading as the dependent variable, and spatial language was used as the mediator. Furthermore, age was included in the model as a covariate (see Figure 3).

Mediation analysis showed that perceptual analogical reasoning was positively associated with spatial language—path a—(*B* = 0.18, *SE* = 0.07, *t* = 2.50, *p* < 0.01, *CI* 95% = [0.035, 0.317]), which in turn was positively associated with map reading—path b—(*B* = 0.43, *SE* = 0.09, *t* = 4.70, *p* < 0.001, *CI* 95% = [0.251, 0.625]). Consequently, the indirect effect of spatial language on the association between perceptual analogical reasoning and map reading was significant (*B*= 0.07, Boot*SE* = 0.03, Boot*CI* 95% = [0.014, 0.152]). Furthermore, the direct effect of perceptual analogical reasoning on map reading—path c—was not significant (*B* = 0.05, *SE* = 0.05, *t* = 1.10, *p* > 0.05, *CI* 95% = [−0.045, 0.156]). These results suggested that spatial language fully mediated the association between perceptual analogical reasoning and map reading. Finally, the total effect—path c’—was significant (*B* = 0.13, *SE* = 0.05, *t* = 2.35, *p* < 0.05, *CI* 95% = [0.019, 0.245]), and the *R^2^* for the entire model was 0.50. Figure 4 summarizes the results of the mediation analysis.

## 4. Discussion

Based on evidence on the role of analogical reasoning and spatial language in human navigation, the present study aimed to further examine the role of perceptual analogical reasoning and map reading and to determine the involvement of spatial language in a sample of typically developing 4- to 6-year-old children. Specifically, we hypothesized that spatial language mediates the association between perceptual analogical reasoning and map reading. After controlling for age, the mediation analysis showed that perceptual analogical reasoning was associated with spatial language, which, in turn, predicted map reading, thereby indicating that spatial language fully mediated the perceptual analogical reasoning–map reading link.

Regarding the association between perceptual analogical reasoning and spatial language (path a of the mediation model), even though some authors have stressed a mutual relationship [26,59], others have argued that analogical competencies are involved in language development mainly in terms of vocabulary and grammar acquisition [29]. Drawing upon the usage-based approach [60,61], the current research suggested that the development of spatial language might depend on analogical reasoning. Particularly, making perceptual analogies allows for the progressive abstraction of spatial linguistic forms through a process of comparison, which leads to the discovery of abstract structural similarities between spatial utterances. Therefore, perceptual analogical reasoning could represent a key process deeply involved in the morphosyntactic component of spatial language, which allows for generating new linguistic forms from previous ones. Regarding the relationship between spatial language and map reading (path b of the mediation model), our results are in line with previous research on spatial abilities in typical and atypical populations. For instance, Hermer-Vasquez et al. [37] showed that preschool children who have acquired spatial locatives, such as left/right, can use a landmark to re-find a hidden object better than children who have not acquired the words. In this vein, Bocchi et al. [42] found that children who comprehended spatial locatives showed better topographic memory. Similarly, preschool children who had learned the term “middle” performed better on a midpoint search task than those who had not [62]. However, research has demonstrated that locative terms and spatial grammar have a central role in navigation ability in individuals with different diseases (i.e., Williams Syndrome and deafness), suggesting that difficulties in spatial language may constrain the development of spatial abilities, including geometrical or landmark information processing [16,63,64]. These findings align with Shusterman and colleagues [65], who argued that spatial expressions allow for semantically combining and coding spatial information about the environment with visual information (e.g., left of the coloured wall), facilitating map reading. Particularly, the mastery of spatial language can serve as a medium for combining information from reasoning systems and use such information to reach a specific navigational goal.

In conclusion, the current study provides evidence that domain-specific factors of language (spatial language) are necessary to apply perceptual analogies and successfully read a map of the surrounding environment.

This research has at least three implications worth mentioning. From a theoretical perspective, it sheds further light on the main mechanisms involved in the association between perceptual analogical reasoning and map reading, providing evidence on the critical role of spatial language. Specifically, our results suggest that spatial language, triggered by perceptual analogical reasoning, can direct children’s attention to, enhance memory for, and integrate different sources of information about the environment [65]. From an educational perspective, this research highlights that spatial language represents a critical mechanism involved in the association between perceptual analogical reasoning and navigational skills in a developmental stage in which the whole children’s cognitive system is developing and not yet mature. This implies that improving spatial language from the early stages of life can be useful in shaping and enhancing children’s map reading across grade levels, thus allowing children to reach higher levels in the ability to read a map and, ultimately, navigational abilities. From a clinical perspective, training in spatial language can be useful in specific neurodevelopmental disorders with deficits in navigation. For instance, training on spatial language could be a useful tool during re-education as a navigation aid for individuals with developmental topographic disorientation (DTD), a disorder that undermines the ability to orient oneself in the environment, learn new routes, and recognize visual scenes [66,67,68,69]. In particular, this training could be useful to prevent the onset of DTD and reduce its social impact.

Furthermore, the current research had a few limitations, which should be considered. First, the study supported the mediating effect of spatial language on the association between perceptual analogical reasoning and map reading using a cross-sectional design, which does not allow for identifying the emergence and development of map-reading abilities. Future research should address this point, stratifying for narrower age groups to monitor when exactly these skills emerge. Additionally, longitudinal research might be fruitful in evaluating time points of map-reading ability and how this ability changes during development, considering the interaction with other abilities that become more complex over time, such as language. Second, this study was conducted enrolling 56 pre-schoolers. Although bootstrapping is a nonparametric approach, which enables a reliable evaluation of the indirect effect in small-sized samples, future research should confirm the mediating effect of spatial language considering a wider sample. Third, we enrolled mainly boys in our research (38 boys); future research should consider a larger and more sex-balanced sample. Fourth, we employed the Semmes Test, which involves the ability to read only simple graphic maps. Since children at 11 months already use information regarding landmarks and landmark arrays [70,71], and given the pivotal role of perceptual processes during the early stages of life [24,25], future research should analyse children’s performance using more complex map-reading tasks in which landmarks and perceptual distractors are reported in the map. Finally, family sociocultural factors were not collected in this research. Given that these variables could play a critical role in shaping children’s language development [72], future research should confirm the mediating role of spatial language while also considering the potential involvement of family influences.

## Figures and Tables

**Figure 1 children-10-00630-f001:**
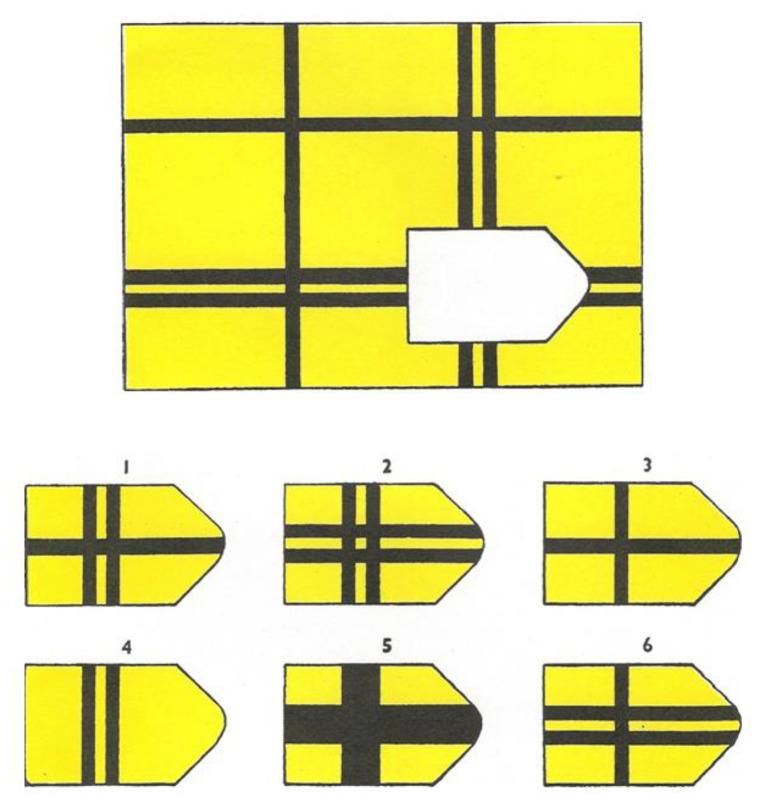
An example of matrix of the Coloured Progressive Matrices.

**Figure 2 children-10-00630-f002:**
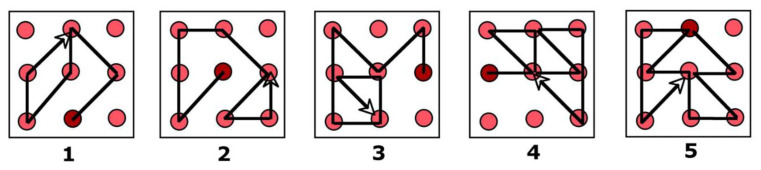
The five maps of the Semmes Test.

**Figure 3 children-10-00630-f003:**
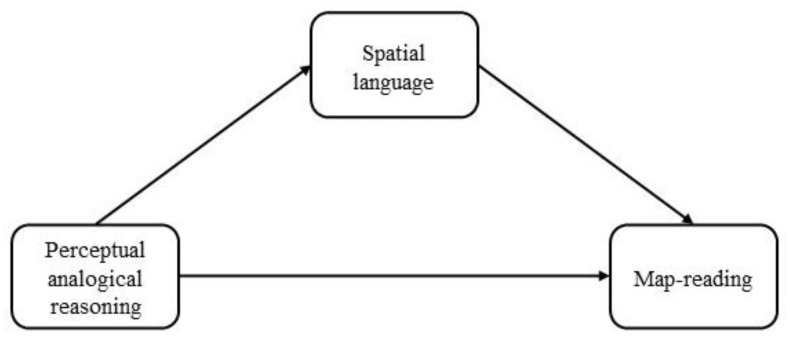
The theoretical mediation model of the current study.

**Figure 4 children-10-00630-f004:**
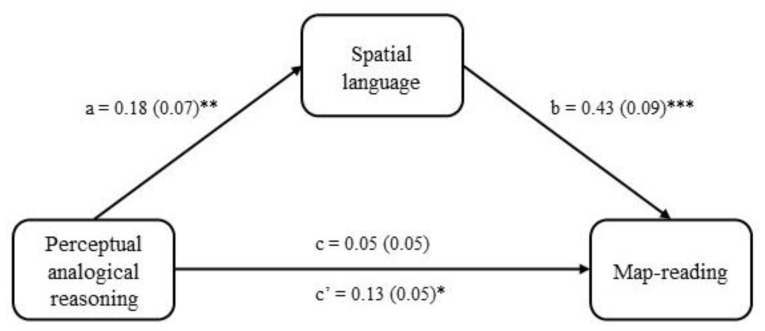
Summary of the results of the mediation model of the current study. The letters a, b, c, and c’ are path coefficients representing unstandardized regression weights and standard errors in parentheses. *** *p* < 0.001, ** *p* < 0.01, * *p* < 0.5.

**Table 1 children-10-00630-t001:** Means, standard deviations, and intercorrelations among all variables.

	M	SD	1	2	3	4	5
1. Age	5.08	0.81	1				
2. Gender			−0.19	1			
3. Perceptual analogical reasoning	17.69	3.74	0.47 **	−0.29 *	1		
4. Spatial language	10.73	1.79	0.15	0.06	0.37 **	1	
5. Map reading	2.94	1.60	0.49 **	−0.06	0.52 **	0.56 **	1

Note. *N* = 56, gender was dummy coded (0 = F; 1 = M), * *p* < 0.05 (two-tailed); ** *p* < 0.01 (two-tailed).

## Data Availability

The dataset will be made available upon request.

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
