# Peer review of "Preschoolers’ Perceptual Analogical Reasoning and Map Reading: A Preliminary Study on the Mediating Effect of Spatial Language"

_children, 2023, doi:10.3390/children10040630_

Round 1

Reviewer 1 Report

Thank you for the opportunity to review this interesting manuscript regarding its topic. The field of analogical reasoning and spatial relations in reading is of vital importance. My major considerations about this manuscript are the following:

1. You evaluated abstract thinking with the Raven test. The age of your participants began from 3 years of age. But Raven measures non-verbal intelligence from 4 to 11 years. You have to explain the modifications and present the appropriate argumentations. Otherwise, the tool is inappropriate for 3year old participants.

2.  You must present how many participants were from the group of 3 years, how many from the group of 4 years, etc. You have different performances according to cognitive maturation due to age level. Also, it is not clear how many boys and girls participated in the current study. You must present a table with the demographic characteristics of the participants. Did you evaluated also the socio-economical status or educational level of parents? 

3. The Procedure is missing. Part of what you present in the Methodology must be in the Procedure. 

4. I have major considerations about the experimental design. You cannot interpret your results with a correlational matrix only. As you have different ages you must apply different analyses for every group. Did you evaluate gender as a contributing factor for the group of participants?

5. In the Discussion part your interpretations need clarification. There is a lack of new citations, and the hypotheses you presented in the Introduction are unanswered.

6. You have to present the Implications of the study as well as the Limitations

Author Response

We want to thank the reviewers for their precious comments and suggestions, which help us improve the overall quality of our manuscript. Below are our responses point by point.

Kind regards, 

The authors

  1. You evaluated abstract thinking with the Raven test. The age of your participants began from 3 years of age. But Raven measures non-verbal intelligence from 4 to 11 years. You have to explain the modifications and present the appropriate argumentations. Otherwise, the tool is inappropriate for 3year old participants.

REPLY: Thank you for this comment. We explain better why we used the Colored Progressive Matrices (CPM) in the paragraph “Method”, stating that “the CPM can be considered as geometric analogy problems, in which the participant is requested to find a specific visual relations system embedded in a graphic matrix by applying analogical reasoning [17,42]. Therefore, in this research, we considered CPM a reliable instrument of perceptual analogical reasoning. This aligns with previous research suggesting that analogical reasoning lies at the heart of visual problem-solving and intelligence more broadly [43]”. Furthermore, we excluded the group of 3 years old children (N=4) from our research. Now, our final sample comprises 56 children (4-6 years old).

  1. You must present how many participants were from the group of 3 years, how many from the group of 4 years, etc. You have different performances according to cognitive maturation due to age level. Also, it is not clear how many boys and girls participated in the current study. You must present a table with the demographic characteristics of the participants. Did you evaluated also the socio-economical status or educational level of parents? 

REPLY: We describe our sample better, stating that “a 56 preschoolers took part in this study (mean age = 5.08 years; SD age = 0.81 years; range age = 4-6 years), whose 38 (67.9%) were boys and the remaining 18 (32.1%) were girls. Of 56 preschoolers, 16 were four years old (28.6%), 19 were five years old (33.9%), and 21 (37.5%) were six years old”. (Please see the paragraph “Participants”). We hope this is enough, and we do not believe providing a Table with the demographic features of the sample is necessary because otherwise, it could be redundant. Finally, we did not evaluate the socio-cultural status of the family, but we included this point as a limitation of our work.

  1. The Procedure is missing. Part of what you present in the Methodology must be in the Procedure. 

REPLY: Fixed, we included the procedure in the manuscript. Please see the paragraph 2.3 “Procedure” (page 5 lines 186-196).

  1. I have major considerations about the experimental design. You cannot interpret your results with a correlational matrix only. As you have different ages you must apply different analyses for every group. Did you evaluate gender as a contributing factor for the group of participants?

REPLY: Please consider that our main focus was the mediating effect of spatial language in the association between perceptual analogical reasoning and map-reading. Given the significant correlation between age and the dependent variable (map-reading), we included age as a covariate in order to control its potential effect in our model. This procedure allowed us to ensure parsimony without performing three mediations for each age group.

  1. In the Discussion part your interpretations need clarification. There is a lack of new citations, and the hypotheses you presented in the Introduction are unanswered.

REPLY: The discussion has been deeply improved in light of our research hypothesis: “spatial language competencies mediate the association between perceptual analogical reasoning and map-reading”. We explained better the mediation, disentangling each significant path of our model.

  1. You have to present the Implications of the study as well as the Limitations

REPLY: We included implications and limitations of our research. Please see pages 7-8.

Reviewer 2 Report

I read with interest the manuscript entitled “Developmental mechanisms in pre-schoolers' map-reading: the contribution of analogical reasoning and spatial language”. In this paper, the authors have assessed the potential mediating effect of spatial language on map reading from abstract reasoning. The aim of the study is interesting, but there are some points that need to improve the overall quality of the manuscript ready for publication.

General

The main problem found is relative to the fact that the so-called analogical reasoning, is in fact the g-factor of intelligence! The authors have to explain why they chose to use Raven’s CPM in order to assess analogical reasoning, while in another paper (DOI: 10.1016/j.tsc.2022.101047)  the authors used another test in order to assess fluid intelligence (which also includes verbal knowledge). 

Moreover, I suggest a general revision of the English language for several mistakes. More in general, proof-reading from native English speakers would be recommended, for a better selection of synonyms and specific terms.

In addition, pay attention to in-text references.

Introduction

In the introduction the word “competence/competencies” is frequently repeated; in some cases (e.g., lines 53, 57) it would be better to replace it with “abilities” (e.g., visuo-spatial abilities). Lines 83-86: the interpretation of the theoretical model proposed by Leroy and colleagues (ref. 26 and 27) is misleading: the authors propose a role of difficulties in analogical reasoning as one of the concurrent causes of problems in building higher-level language schemas and generalization of grammatical rules. As it is proposed in the text, it seems that analogical reasoning’s difficulties are the only cause of SLI.
Line 97-98: the sentence states that “Further evidence of the relationship between spatial language and spatial ability, as navigational ability…”, is it possible that the authors were referring to spatial awareness? In the following sentences the authors proceed with the definition of spatial language and spatial awareness, but they cited only the first in the previous paragraph.

Material and methods

It is better to report the ethics committee approval and informed consent authorization also in the participants section. Additionally, it is important to provide the number and date of authorisation.

Please report here the results of a power analysis aimed at defining the minimum sample size needed for the planned analyses. G*power software, R or online calculators will be useful at this point.  See also DOI: 10.1111/j.1467-9280.2007.01882.x. Alternatively you can report the sensitivity analysis. 

A better description of the spatial navigation task should be provided; as it is presented in the paper, the reader is forced to retrieve the original paper in which the task was first presented in order to better understand administration procedures. 

Results

Lines 184-185. I would appreciate it if you could report each test's results and their significance separately. It is difficult to read Z-raven etc.

I understand the process of calculating the mediating effects, but the basic principles of presenting mediating analyses will be respected. I’m referring to the lack of reporting step by step all the different outcomes in the right way. This is because regressions needed to confirm the efficacy of the mediation model. See the referenced paper for details of the four steps. Bivariate and multiple regressions must be reported including Beta, CI and significance.

Lines 195-196, it’s better to call independent, mediator and dependent variables instead of focal and outcome.

Line 196, the model number is useless or it needs to be explained.

It will be useful to report coefficients in the figure, as is the case in many publications. E.g. DOI: 10.1080/10400419.2022.2092338

Discussion

Discussion focuses on the importance of investigating, in general, the relevance of spatial navigation abilities and does not mention the relevance of the present findings in light of existing literature. It seems a general discussion and presentation of existing literature, such as a copy of introduction, but does not highlight the connection with the findings presented in the paper.

Author Response

We want to thank the reviewers for their precious comments and suggestions, which help us improve the overall quality of our manuscript. Below are our responses point by point.

Kind regards, 

The authors

 General

The main problem found is relative to the fact that the so-called analogical reasoning, is in fact the g-factor of intelligence! The authors have to explain why they chose to use Raven’s CPM in order to assess analogical reasoning, while in another paper (DOI: 10.1016/j.tsc.2022.101047)  the authors used another test in order to assess fluid intelligence (which also includes verbal knowledge). 

Moreover, I suggest a general revision of the English language for several mistakes. More in general, proof-reading from native English speakers would be recommended, for a better selection of synonyms and specific terms.

In addition, pay attention to in-text references.

REPLY: We clarified this point by stating that Colored Progressive Matrices (CPM) can be considered geometric analogy problems, in which the participant is requested to find a specific visual relations system embedded in graphic matrices by applying analogical reasoning (Kunda et al., 2012). This implies that CPM represents a reliable index of perceptual analogical reasoning. Furthermore, note that this logic aligns with previous research suggesting that analogical reasoning lies at the heart of visual problem solving and intelligence more broadly (Lovett & Forbus, 2017). All information about this point is included in the description of the CPM.

Furthermore, we revised the English and carefully checked the references throughout the manuscripts.

Introduction

In the introduction the word “competence/competencies” is frequently repeated; in some cases (e.g., lines 53, 57) it would be better to replace it with “abilities” (e.g., visuo-spatial abilities).

REPLY: Fixed.

Lines 83-86: the interpretation of the theoretical model proposed by Leroy and colleagues (ref. 26 and 27) is misleading: the authors propose a role of difficulties in analogical reasoning as one of the concurrent causes of problems in building higher-level language schemas and generalization of grammatical rules. As it is proposed in the text, it seems that analogical reasoning’s difficulties are the only cause of SLI.

REPLY: We improved this section of the paragraph, stating that “Leroy and colleagues [24,25] argued that the deficit in linguistic and non-linguistic (perceptual) analogical reasoning represents one of the leading causes of language disorders in children with Specific Language Impairment. Overall, these findings offer theoretical grounds to reasonably assume an essential role of analogical skills in language capabilities”.  

Line 97-98: the sentence states that “Further evidence of the relationship between spatial language and spatial ability, as navigational ability…”, is it possible that the authors were referring to spatial awareness? In the following sentences the authors proceed with the definition of spatial language and spatial awareness, but they cited only the first in the previous paragraph.

REPLY: We dropped the sentence and the citation. However, we clarified better this point, providing a description of Piccardi and colleagues’ research [37], in which the authors found that “children’s verbal comprehension of spatial locatives (e.g., in front of/behind, from/to, below/above, up/down, etc.) predicted the ability to learn, retrieve, and graphically represent pathways on a map of the environment”.

Material and methods

It is better to report the ethics committee approval and informed consent authorization also in the participants section. Additionally, it is important to provide the number and date of authorisation.

Please report here the results of a power analysis aimed at defining the minimum sample size needed for the planned analyses. G*power software, R or online calculators will be useful at this point.  See also DOI: 10.1111/j.1467-9280.2007.01882.x. Alternatively you can report the sensitivity analysis. 

REPLY: Unfortunately, we did not perform an a priori power analysis. However, note that we explained in the paragraph “Statistics” that bootstrapping is a non-parametric method, which allows for finding an accurate evaluation of the indirect effect also in small and medium samples (please, see page 5, lines 204-207). Furthermore, note that in order to verify if our model was underpowered, we performed a G*Power post-hoc power analysis. We reported here the power value (0.98), which satisfied the recommended cut-off value of 0.80 (Cohen, 1992). Finally, please, note that the small sample size was included as a limitation of our research (please see page 8, lines 315-317).  

Cohen, J. (1992). A power primer. Psychological Bulletin, 122(1), 155–159.

A better description of the spatial navigation task should be provided; as it is presented in the paper, the reader is forced to retrieve the original paper in which the task was first presented in order to better understand administration procedures. 

REPLY: We provided a better and more comprehensive description of the Semmes Test. Please note that we also improved the descriptions of the other measures used in our research.

Results

Lines 184-185. I would appreciate it if you could report each test's results and their significance separately. It is difficult to read Z-raven etc.

REPLY: Fixed.

I understand the process of calculating the mediating effects, but the basic principles of presenting mediating analyses will be respected. I’m referring to the lack of reporting step by step all the different outcomes in the right way. This is because regressions needed to confirm the efficacy of the mediation model. See the referenced paper for details of the four steps. Bivariate and multiple regressions must be reported including Beta, CI and significance.

REPLY: We improved the section of the mediation analysis, reporting results as follows “Mediation analysis showed that perceptual analogical reasoning was positively associated with spatial language – path a – (B= 0.18, SE = 0.07, t = 2.50, p < 0.01, CI 95% = [0.035, 0.317]), which in turn was positively associated with map-reading – path b - (B= 0.43, SE = 0.09, t = 4.70, p < 0.001, CI 95% = [0.251, 0.625]). Consequently, the indirect effect of spatial language on the association between perceptual analogical reasoning and ma-reading was significant (B= 0.07, BootSE = 0.03, BootCI 95% = [0.014, 0.152]). Furthermore, the direct effect of perceptual analogical reasoning on map-reading – path c – was not significant (B= 0.05, SE = 0.05, t = 1.10, p > 0.05, CI 95% = [-0.045, 0.156]). These results suggested that spatial language fully mediated the association between perceptual analogical reasoning and map-reading. Finally, the total effect – path c’ – was significant (B= 0.13, SE = 0.05, t = 2.35, p < 0.05, CI 95% = [0.019, 0.245]), and the R2 for the entire model was 0.50. Figure 4 summarizes the results of the mediation analysis”

Lines 195-196, it’s better to call independent, mediator and dependent variables instead of focal and outcome.

REPLY: Fixed.

Line 196, the model number is useless or it needs to be explained.

REPLY: We dropped the number of the model in the section.

It will be useful to report coefficients in the figure, as is the case in many publications. E.g. DOI: 10.1080/10400419.2022.2092338

 REPLY: We included in the manuscript an additional figure (Figure 4) in which we summarized the results of the mediation analysis, including all paths with regression weights and standard errors in parenthesis. 

Discussion

Discussion focuses on the importance of investigating, in general, the relevance of spatial navigation abilities and does not mention the relevance of the present findings in light of existing literature. It seems a general discussion and presentation of existing literature, such as a copy of introduction, but does not highlight the connection with the findings presented in the paper.

REPLY: We revised the discussion, explaining our results in light of our research hypothesis, “spatial language competencies mediate the association between perceptual analogical reasoning and map-reading”. Specifically, we discuss our results focusing on the significant paths of our mediation model. Moreover, we described the implications of our findings and potential limitations. 

Round 2

Reviewer 1 Report

The authors have followed all the reviewer's recommendations and corrected their manuscript. 

Reviewer 2 Report

The authors have made all modification needed and the overall quality of the manuscript was improved.